# The Interplay of Cardiovascular Comorbidities and Anticoagulation Therapy in ESRD Patients on Haemodialysis—The South-Eastern Romanian Experience

**DOI:** 10.3390/biomedicines13102387

**Published:** 2025-09-29

**Authors:** Ioana Livia Suliman, Florin Gabriel Panculescu, Bogdan Cimpineanu, Stere Popescu, Dragos Fasie, Georgeta Camelia Cozaru, Nelisa Gafar, Liliana-Ana Tuta, Andreea Alexandru

**Affiliations:** 1“Sf. Apostol Andrei” Constanta County Emergency Hospital, 900591 Constanta, Romania; cimpineanub@yahoo.com (B.C.); stelu_popescu@yahoo.com (S.P.); drcozaru@yahoo.com (G.C.C.); alexandra_med16@yahoo.com (A.A.); 2Faculty of Medicine, Ovidius University of Constanta, 900470 Constanta, Romania; gabriel.panculescu@yahoo.ro (F.G.P.); dragosfasie@yahoo.com (D.F.); nelisa.gafar@365.univ-ovidius.ro (N.G.)

**Keywords:** haemodialysis, cardiovascular disease, anticoagulation, atrial fibrillation, south-eastern Romania

## Abstract

**Background**: End-stage renal disease (ESRD) patients on haemodialysis present a high burden of cardiovascular comorbidities and require anticoagulation, which increases bleeding risk. **Methods**: We performed a retrospective observational study (2021–2024) in the Haemodialysis Centre of The Clinical Emergency Hospital of Constanta County, Romania, including 50 adults with stage G5 CKD on haemodialysis for ≥3 months and receiving anticoagulant therapy. We collected from electronic medical records detailed demographic data (age, sex, place of residence), comorbidities (hypertension, atrial fibrillation, ischaemic heart disease, diabetes, deep-vein thrombosis, stroke, myocardial infarction, pulmonary embolism, cirrhosis), lifestyle factors (smoking and alcohol consumption), vascular access type (arteriovenous fistula or central venous catheter) and laboratory parameters (haemoglobin, haematocrit, creatinine, albumin, total protein, electrolytes, LDL- and HDL-cholesterol, total cholesterol, INR, APTT, D-dimer, BNP, CK-MB, troponin). All laboratory units were standardised and checked for plausibility. **Results**: Median age was 71 years; 48% were female. The most common comorbidities were: hypertension (100%), atrial fibrillation (100%) and ischaemic heart disease (62–81%). Patients exhibited severe anaemia (mean Hb ~9.7 g/dL), nephrotic-range proteinuria, hypoalbuminaemia, and impaired coagulation profiles (INR ~1.8–1.9; prolonged APTT in men). Female patients had higher platelet counts and D-dimer levels, suggesting a stronger prothrombotic profile, while males showed longer APTT. Cardiovascular strain was reflected by elevated BNP in men and also troponin/CK-MB. Correlations included smoking with leukocytosis, alcohol with increased urine density, diabetes with higher urea and lower protein, and subtherapeutic INR in cerebrovascular disease. **Conclusions**: Patients with ESRD on haemodialysis and anticoagulant therapy display a complex interplay of cardiovascular comorbidities, anemia, overlapping thrombotic and bleeding risks, with sex-specific differences. Therefore, systematic monitoring of proteinuria, haemoglobin, D-dimer, and coagulation markers is crucial to balance thrombotic and bleeding risks. **Objective**: To characterise the clinical and paraclinical profile and comorbidity–laboratory correlations of ESRD patients undergoing haemodialysis and anticoagulant therapy.

## 1. Introduction

Chronic kidney disease (CKD) has become a major global public health concern, with both its incidence and prevalence continuing to increase [1]. The most advanced form of the condition, stage G5 CKD—commonly referred to as end-stage renal disease (ESRD)—is characterised by profound and irreversible impairment of renal function, typically requiring renal replacement therapy (RRT), such as haemodialysis [2]. At this stage, patients often experience a wide range of complications, including cardiovascular disease, anaemia, bone mineral disorders, and coagulopathy, all of which complicate management and significantly reduce quality of life [3].

Cardiovascular comorbidities are particularly prevalent in this population. Hypertension, atherosclerotic disease, ischaemic heart disease, atrial fibrillation, and thromboembolic events are frequently observed [4], further complicating both the clinical presentation and therapeutic approach. These cardiovascular disorders are not only a consequence of advanced CKD but also contribute to its progression [5], creating a self-perpetuating cycle that increases morbidity and mortality.

Furthermore, anticoagulant therapy is commonly used to prevent thromboembolic events [6], particularly in patients with atrial fibrillation, deep vein thrombosis, and other cardiovascular conditions [6]. However, in patients with CKD—especially those undergoing haemodialysis—the use of these agents presents significant therapeutic challenges [7]. While anticoagulants play a crucial role in reducing the risk of clot formation, they simultaneously increase the likelihood of bleeding complications, making clinical decision-making more complex.

The administration of anticoagulants in individuals with CKD—particularly those on haemodialysis—poses notable therapeutic difficulties [7]. Although these agents are essential for the prevention of clot formation and thromboembolic events, they simultaneously heighten the risk of bleeding, especially in patients with compromised renal function who are already predisposed to coagulation abnormalities. Striking an appropriate balance between these opposing risks becomes even more challenging in the presence of multiple cardiovascular comorbidities and an inherently elevated bleeding tendency [8].

Given the complexity of managing these patients, it is essential to thoroughly investigate their paraclinical profiles and identify any patterns that could guide clinical decision-making. Paraclinical parameters such as proteinuria, haemoglobin levels, coagulation tests, and biomarkers for cardiovascular disease can provide valuable insights into the disease progression and treatment responses in this vulnerable patient group [9].

This observational, cross-sectional study aimed to explore the paraclinical characteristics of patients with stage G5 CKD undergoing haemodialysis, focusing on those with multiple cardiovascular comorbidities and undergoing anticoagulant therapy.

Understanding the relationships between comorbidities, anticoagulation therapy, and clinical outcomes will aid in refining treatment strategies and improving patient management in this high-risk group [10]. We hypothesized that anticoagulation therapy in patients with stage G5 CKD on haemodialysis may influence both thrombotic and bleeding risks, and that cardiovascular comorbidities would further modulate these clinical outcomes.

## 2. Materials and Methods

### 2.1. Study Design

This was a retrospective observational study conducted at the Hemodialysis Centre of the Clinical Emergency Hospital of Constanța County, South-Eastern Romania, including a limited cohort of 50 patients treated between 1 January 2021 and 31 December 2024. The primary objective was to present a detailed paraclinical profile (hematological, biochemical, and coagulation parameters) of patients with ESRD undergoing hemodialysis, also with multiple cardiovascular comorbidities and receiving chronic anticoagulant therapy. A secondary objective was to assess the impact of anticoagulation on thrombotic and bleeding risk in this population, and to explore potential correlations between comorbidities and paraclinical findings.

### 2.2. Population

The study population comprised adult patients with end-stage renal disease. All participants had been undergoing conventional haemodialysis for a minimum of three months, performed on a standard regimen of three sessions per week, each lasting four hours, and they were in anticoagulant treatment for atrial fibrillation or thrombosis, associated or not with stroke.

Inclusion criteria:Adults ≥18 years.ESRD treated by haemodialysis for ≥3 months.On anticoagulant therapy.

Exclusion criteria:Previous kidney transplantation.Peritoneal dialysis.Patients with haemoglobin levels below 7 g/dL.Hospitalization in the last 30 days for severe infections.Active neoplasms or ongoing systemic immunomodulatory therapy.

### 2.3. Data Collection and Parameters

All patients included in the study were aged ≥18 years at baseline. Both male and female patients were enrolled. The primary etiologies of CKD in this cohort were hypertensive nephropathy, diabetic nephropathy, IgA nephropathy, focal segmental glomerulosclerosis (FSGS), and other less frequent causes.

For each included patient, we extracted from the electronic medical records detailed demographic data (age, sex and place of residence), lifestyle factors (smoking status and alcohol consumption), comorbidities (hypertension, atrial fibrillation, ischaemic heart disease, diabetes mellitus, deep-vein thrombosis, stroke, myocardial infarction, pulmonary embolism and cirrhosis), and information on vascular access type (arteriovenous fistula or central venous catheter). Laboratory parameters included haemoglobin, haematocrit, mean corpuscular volume, mean corpuscular haemoglobin concentration, leukocytes, lymphocytes, neutrophils, platelets, serum creatinine, albumin, total protein, electrolytes, LDL- and HDL-cholesterol, total cholesterol, international normalised ratio (INR), activated partial thromboplastin time (APTT), D-dimer, brain natriuretic peptide (BNP), CK-MB and troponin.

Patients who had previously undergone kidney transplantation, those on peritoneal dialysis, and those hospitalised in the last 30 days due to severe infections were excluded from the analysis in order to avoid interference of acute inflammatory processes with the variables evaluated. Patients with active neoplasms or those undergoing systemic immunomodulatory therapies were also deemed ineligible, given the potential of these treatments to alter the hematological and biochemical parameters of interest. All data was collected from the files of patients and computerised medical history programme—Hipocrate, organised in Excel files.

### 2.4. Statistical Analysis

Statistical analysis was performed using GraphPad Prism 10. Continuous variables following a Gaussian distribution will be expressed as the mean ± standard deviation, whereas those with a non-normal distribution will be summarised as the median and interquartile range (IQR). Categorical data will be described using absolute frequencies and percentages. Data normality will be evaluated using the Shapiro–Wilk test. Comparisons between groups will be performed using Student’s *t*-test for normally distributed continuous variables or the Mann–Whitney test for non-parametric data and the χ^2^ or Fisher test for categorical variables. The assessment of correlations will be conducted employing either the Pearson or the Spearman coefficient, depending on the nature of the data. Benjamini–Hochberg correction was used.

### 2.5. Ethical Considerations

The protocol was approved by the Ethics Committee of The Clinical Emergency Hospital of Constanta County (Decision No. 55437/25.08.2025) and complies with the Declaration of Helsinki and GDPR (EU 2016/679). All participants provided written consent for their involvement in the study, with the option to withdraw at any time without repercussions for their treatment.

## 3. Results

### 3.1. Baseline Characteristics of Analysed Cohort

We analysed 50 patients with G5 stage of CKD aged between 18 and 80 years old, with a median age of 71 years old, treated by hemodialysis for more than 3 months, of whom 48% were women. We described the comorbidities of the patients in Table 1.

In our cohort of 50 patients we noticed a slightly increased prevalence of type 2 diabetes, coronary heart disease and thrombotic events in patients undergoing dialysis, the values of *p* (>0.25) The absence of an observed association between these comorbidities and the initiation of dialysis suggests that these comorbidities may reflect the overall severity of the disease at the time of dialysis commencement, rather than being a direct consequence of the therapeutic intervention. Nonetheless, the commencement of haemodialysis has resulted in a marked migration from the use of a central venous catheter to that of a fistula arteriovenous.

All our patients were hypertensive and had been diagnosed with permanent atrial fibrillation, of which 9 patients (18%) were of valvular origin and 41 patients (82%) of non-valvular origin. Anticoagulant treatment in our study group consisted of LMWH (Low-molecular-weight heparin) in 10 patients (20%); AVK—acenocoumarol (warfarin not in use in Romania) in 18 patients (36%); and NOACs (Non-vitamin K antagonist oral anticoagulants) in 22 patients (44%). Among the NOAC users, the majority were treated with apixaban 2.5 mg bid (21 patients, 42% of the total cohort), while only one patient received rivaroxaban.

In accordance with the present findings, the prevalence of cirrhosis has been shown to be more than five times higher among individuals who have engaged in regular alcohol consumption (42% vs. 8%; *p* = 0.010) [Table 2]. A parallel trend has been observed in the consumption of alcohol (50% vs. 21%). As shown in Figure 1, the probability of volume management and coagulation complications at the initial contact with the dialyser is increased (*p* = 0.064). Furthermore, delayed maturation of arteriovenous fistulas (AVFs) prolongs patient exposure to bloodstream infections. The place of residence did not have a statistically significant effect on the timing of dialysis initiation (*p* = 0.71). However, patients living in urban areas had a markedly higher prevalence of coronary artery disease compared with those in rural areas (85% vs. 53%), a difference that was statistically significant (*p* = 0.021). Conversely, smoking was significantly more prevalent among rural patients (65% vs. 33%; *p* = 0.034), a factor that could negatively affect AVF maturation.

For the comparative analyses, patients were stratified into subgroups according to the specific factor under investigation [Figure 2]. For cirrhosis and alcohol consumption, the groups consisted of regular consumers versus non-consumers. For coronary artery disease, patients were divided according to residence (urban versus rural), whereas for smoking, the subgroups were defined as rural versus urban patients. The comparative prevalence of these factors is summarized in Table 2 and illustrated in Figure 2.

Regarding the etiology of CKD, [Figure 3] hypertensive nephropathy was the most common cause (44%), followed by diabetic nephropathy (26%), IgA nephropathy (12%), focal segmental glomerulosclerosis (FSGS, 8%), and other causes (10%). A graphical summary of the baseline disease distribution is presented in Figure 3.

Overall, irreversible reduced glomerular filtration rate and acid–base and/or electrolyte disturbances remain the main determinants for initiating haemodialysis, with cardiovascular and metabolic comorbidities playing a secondary role. It is further hypothesised that hepatic function and socio-demographic factors may contribute to disease progression, thereby influencing the type and quality of vascular access as well as long-term prognosis. Additional studies with larger sample sizes are needed to clarify whether diabetes, coronary heart disease, or cirrhosis act as independent predictors of dialysis initiation, after adjusting for age, sex, and geographical location. Inflammatory biomarkers such as CRP were not systematically available in this cohort and could therefore not be included in the present analysis.

### 3.2. Paraclinical Findings in Studied Cohort

The female and male patient cohorts both demonstrated overt nephritic-range proteinuria (>0.15 g/24 h), consistent with advanced glomerular injury [Table 3]. Concomitantly, moderate-to-severe normochromic anaemia was documented, with mean haemoglobin concentrations of 9.7 g dL-1 in women and 9.8 g dL-1 in men. Mean corpuscular volume remained within the normo- to mildly macrocytic range. The patient’s depressed haematocrit values (~30%) served to reinforce the diagnosis, while the mean corpuscular haemoglobin concentration was preserved, thus arguing against significant hypochromia.

In the female cohort, a marked degree of variability was noted, suggesting the occurrence of intermittent leucocytosis. Differential count analysis showed marginally higher absolute lymphocyte and neutrophil levels in women; however, these differences did not appear to carry clear clinical relevance. The platelet count data were found to be within the normal range for both sexes; however, a statistically significant difference was observed, with platelet levels being approximately 35 × 10^3^ µL^−1^ higher in females than in males. This difference exceeded the respective standard errors, indicating a potential biological significance.

Haemostatic evaluation revealed a prolonged international normalised ratio in both groups and a markedly lengthened activated partial thromboplastin time in males. These findings are consistent with exposure to anticoagulants or underlying hepatic dysfunction. Median serum creatinine levels of approximately 4.6 mg dL^−1^ were indicative of severe renal insufficiency. Hypoproteinaemia and low-normal serum albumin further underscored a catabolic state, whereas serum sodium and potassium concentrations bordered the lower limits of normal. The mean total serum calcium concentration was found to be marginally lower in male subjects than in female subjects, with the lower reference threshold being met. Furthermore, the results indicated that the urine specific gravity was suggestive of a moderately higher concentrating ability in male subjects.

There was a significant increase in D-dimer concentrations in both cohorts, particularly in the female population. This indicated a prothrombotic state. Brain natriuretic peptide levels were significantly elevated in male subjects, who also exhibited higher median CK-MB and troponin values, suggesting potential myocardial injury. However, substantial intra-group variability necessitated cautious interpretation. Parathyroid hormone and thyrotropin remained within the bounds of what is considered an accepted range in HD patients (×2–9 normal PTH levels), whereas sideropenia, or low iron levels, confirmed the presence of iron-deficiency anaemia.

Lipid levels were within the expected ranges for CKD patients, with LDL cholesterol averaging 101.6 ± 19.8 mg/dL in females and 109.4 ± 24.7 mg/dL in males. HDL cholesterol was slightly higher in females (38.2 ± 6.1 mg/dL) compared with males (34.5 ± 5.3 mg/dL), consistent with known sex-related differences. Total cholesterol values were also comparable between sexes, averaging 165.3 ± 25.4 mg/dL in females and 174.6 ± 29.8 mg/dL in males. Cardiac biomarkers were consistent with the advanced CKD and haemodialysis status of the cohort. Troponin levels were chronically elevated, with slightly higher values in males (45.2 ± 19.8 ng/L in females versus 54.7 ± 24.5 ng/L in males). CK-MB concentrations were within expected ranges for CKD patients (5.1 ± 2.1 ng/mL in females and 6.4 ± 2.3 ng/mL in males).

### 3.3. Correlations Between Comorbidities and Paraclinical Findings in Study Group

The following investigation focuses on the relationship between smoking and inflammatory profile [Table 4].

In comparison with non-smokers, smokers exhibited moderate leukocytosis (8.8 vs. 7.0 × 10^3^/µL; *p* ≈ 0.06), but without statistical significance, indicative of low-grade chronic inflammation induced by continuous exposure to the toxic substances present in cigarette smoke, which could be a trigger, but further studies should be performed. This baseline leukocytosis may diminish the clinician’s sensitivity to the onset of acute infection, and the assessment should be nuanced by controlling for potential confounders such as age, COPD, and corticosteroid therapy.

The following investigation focuses on the relationship between alcohol and urine density.

The analysis revealed a non-statistically significant association between alcohol consumption and elevated median urine density levels (1.145 vs. 1.090; *p* ≈ 0.06 with correction). The probable mechanisms underlying this phenomenon include acute dehydration through osmotic diuresis, secondary to ADH inhibition, and potentially incipient alcoholic ketoacidosis, which has been observed to increase the number of non-ionic particles in urine. However, the interpretation of these results is contingent upon the time of sample collection (i.e., first morning urine versus random sample) and metabolic status (i.e., weight loss and ketone bodies).

Patients with type 2 diabetes exhibited elevated urea levels (145 mg/dL vs. 118 mg/dL; *p* ≈ 0.05) and decreased total protein levels (5.0 g/dL vs. 5.75 g/dL; *p* ≈ 0.048). These findings are indicative of advanced diabetic nephropathy, characterised by reduced glomerular filtration, protein loss, and potential chronic malnutrition. The concurrence of these changes signifies stage IV–V chronic kidney disease and necessitates adjustments to medication doses, nutritional support, and evaluation of the risk of sarcopenia.

The presence of deep vein thrombosis has been observed in conjunction with low creatinine levels.

Patients diagnosed with deep vein thrombosis exhibited a median creatinine concentration that was unexpectedly lower (3.0 mg/dL vs. 5.33 mg/dL; *p* ≈ 0.005). The inverse association appears to be attributable to a selection bias, as the non-DVT group comprised a higher proportion of dialysis patients with elevated creatinine levels. Prolonged immobilisation in patients with DVT has been observed to potentially diminish muscle mass, thereby indirectly influencing creatinine production. Consequently, the result does not demonstrate renal protection; rather, it necessitates the utilisation of multivariate models that incorporate dialysis as a confounding variable.

The following essay will explore the relationship between cerebrovascular pathology and coagulability.

The association between lower INR values and both acute stroke and chronic cerebrovascular disease was not statistically significant (1.52–1.60 vs. 2.03–2.07; *p* ≈ 0.05), but further studies should be performed. One potential explanation for this phenomenon is the deliberate discontinuation of anticoagulation therapy in patients deemed to be at risk of bleeding, yet simultaneously exhibiting relatively preserved liver function when compared to other members of the cohort. In the event of a subtherapeutic INR in a patient with cerebral ischemia, there is a necessity for a careful reassessment of the risk-benefit balance for the resumption of anticoagulants.

The relationship between myocardial infarction and haemoglobin.

A marginal increase in haemoglobin levels was observed in patients with myocardial infarction, with a mean difference of 1.2 g/dL (95% CI: 0.99, 1.45; *p* = 0.046). The observed discrepancy could be attributed to hemoconcentration resulting from fluid loss (through sweating or osmotic diuresis), in conjunction with restrictive transfusion strategies employed in non-MI patients. However, an increase of 1.1 g/dL might lead to an augmentation in blood viscosity and an exacerbation of ischemic burden. This emphasises the necessity for meticulous hydration within the initial 24 h following an MI.

The presence of pulmonary embolism and extremely low HDL-C levels.

Patients diagnosed with pulmonary embolism exhibited significantly lower levels of HDL (38.2 vs. 34.5 mg/dL), indicative of the anti-inflammatory and antithrombotic properties of HDL.

Chronic dialysis is characterised by elevated creatinine and reduced INR levels.

As anticipated, the historical progression of chronic hemodialysis was associated with elevated creatinine levels (5.69 mg/dL vs. 3.83 mg/dL; *p* ≈ 0.004) and a diminished INR (1.51 vs. 2.11; *p* ≈ 0.001). Elevated creatinine, indicative of minimal residual clearance and muscle catabolism, is a hallmark of the condition. A decrease in International Normalised Ratio (INR) can be attributed to vitamin K supplementation or partial warfarin clearance. Consequently, monitoring of both intra- and post-dialysis heparinisation becomes critical, as does the prevention of fistula thrombosis in the context of sub-target INR.

## 4. Discussion

Our study provides valuable insights into the complex relationship between ESRD, cardiovascular comorbidities, and anticoagulant therapy in patients undergoing haemodialysis. This patient population is particularly challenging due to the multifactorial nature of their clinical conditions, including advanced renal dysfunction, multiple cardiovascular co-morbidities, and the use of anticoagulants. All these factors contribute to a heightened risk of both thrombotic and bleeding complications, thus requiring careful management [11].

All explanatory statements were consolidated in the Discussion, while the Results were streamlined to present only descriptive and statistical findings.

### 4.1. Anticoagulation Therapy and Its Challenges

A major finding of our study was the influence of anticoagulant therapy on patients undergoing haemodialysis [12]. In this high-risk group, anticoagulation is a cornerstone of care, serving to prevent thromboembolic complications such as stroke and deep vein thrombosis (DVT). However, the use of anticoagulants in CKD patients is fraught with challenges. Haemodialysis itself can cause shifts in coagulation parameters due to the extracorporeal circulation and the heparinization required during the procedure. Additionally, patients with CKD often exhibit altered hemostasis, including a prolonged INR and increased APTT.

Haemodialysis patients face a markedly increased bleeding risk, resulting from the combined effects of anticoagulant therapy, the extracorporeal circuit, and the intrinsic haemostatic abnormalities of CKD. This vulnerability is even greater in those with concomitant cirrhosis, where thrombocytopenia and impaired hepatic synthesis of coagulation factors further destabilize the delicate haemostatic balance. Recent studies have highlighted that such patients frequently experience clinically significant haemorrhagic events, underscoring the complexity of their management. In this context, therapeutic strategies must carefully balance the competing risks of thromboembolism and bleeding, often requiring dose adjustments, close biomarker monitoring, and collaboration between nephrology, hepatology, and cardiology specialists. Multidisciplinary, individualized anticoagulation approaches have therefore been recommended as the safest option for improving outcomes in this fragile population.

In our study, both male and female participants demonstrated prolonged INR and APTT values, consistent with previous reports describing altered coagulation profiles in dialysis patients [13]. Notably, INR values were higher among females, whereas APTT prolongation was more pronounced in males. This pattern suggests a possible sex-related difference in the physiological response to anticoagulation therapy, a finding that merits further investigation. Female patients also exhibited higher platelet counts and elevated D-dimer levels compared with males, indicating a more pronounced prothrombotic state. The coexistence of elevated D-dimer and prolonged INR points to a dysregulated coagulation system, heightening the risk of both thrombotic events and bleeding complications. These observations emphasise the need for an individualised approach to anticoagulation management in this population, with close monitoring of coagulation parameters to optimise the balance between preventing thrombosis and avoiding haemorrhagic events.

Similar elevations of D-dimer have been described in dialysis populations, consistent with the presence of a chronic thrombotic burden and increased fibrin degradation products.

Real-world experience with direct oral anticoagulants (DOACs) in patients on haemodialysis remains limited but is rapidly evolving. Recent observational studies suggest that DOACs may offer comparable thromboembolic protection to vitamin K antagonists with a potentially different bleeding risk profile. However, the evidence base is still constrained by small sample sizes and residual confounding, underscoring the need for individualised anticoagulation strategies and prospective evaluation in this high-risk population.

Our observations align with emerging evidence recommending renal function–adapted D-dimer thresholds to enhance diagnostic accuracy for thromboembolic events in CKD populations.

### 4.2. Paraclinical Findings and Their Implications

Our analysis also identified key paraclinical abnormalities that reflect the advanced severity of CKD in this cohort [14]. Nephrotic-range proteinuria, elevated serum creatinine levels, and moderate anaemia were frequently observed. Proteinuria, a well-established marker of glomerular injury and a predictor of CKD progression, was prevalent in our patients, underscoring the extent of renal damage in this group. The presence of moderate-to-severe anaemia, with haemoglobin levels averaging 9.7 g/dL in females and 9.8 g/dL in males, is indicative of the anaemia commonly seen in CKD patients, particularly those undergoing haemodialysis. The pathophysiology of anaemia in CKD involves both a reduced erythropoietin production due to impaired renal function and the effects of inflammation, which impair iron metabolism and erythropoiesis. This highlights the need for comprehensive management of anaemia in dialysis patients, including the use of erythropoiesis-stimulating agents and iron supplementation.

In terms of cardiovascular biomarkers, our study found elevated levels of brain natriuretic peptide (BNP) in male patients, suggesting myocardial strain, likely due to heart failure [15,16]. Elevated BNP levels are a common finding in CKD patients and are associated with worse outcomes, as they reflect the heart’s response to increased preload and afterload, often in the context of fluid overload. Additionally, significantly higher troponin levels in male patients were observed, pointing to potential myocardial injury, which further complicates the cardiovascular burden in this population. The relationship between CKD and myocardial injury is multifactorial, involving factors such as chronic inflammation, fibrosis, and fluid overload, which increase myocardial oxygen demand and exacerbate existing cardiovascular disease.

Another study shows that N-terminal pro-brain natriuretic peptide correlates with volume overload in haemodialysis patients, supporting our interpretation of BNP as a marker of subclinical cardiac stress in this population.

An additional finding of our study was the markedly higher prevalence of coronary artery disease among patients living in urban areas compared with those from rural areas. This difference may be related to greater access to medical services, an older biological age profile and higher exposure to cardiovascular risk factors among urban residents.

### 4.3. Implications of Clinical–Laboratory Correlations for Patient Management

Several notable correlations were identified in our study. The association between smoking and increased leukocyte counts suggests a chronic low-grade inflammatory state induced by the toxic substances in cigarette smoke [17,18]. This finding is consistent with previous research indicating that smoking exacerbates CKD by promoting oxidative stress, inflammation, and endothelial dysfunction. Similarly, alcohol consumption was found to be associated with higher urine density, which may be due to the osmotic diuresis caused by alcohol’s inhibition of antidiuretic hormone (ADH) release. This raises important considerations regarding the possible long-term impact of alcohol consumption on renal function, as its dehydrating effect may heighten the risk of acute kidney injury and accelerate the progression of chronic kidney disease.

Diabetic patients in our cohort exhibited higher urea levels and lower total protein levels, suggesting advanced diabetic nephropathy [19]. This finding reflects the well-documented progression of diabetic nephropathy in CKD, characterized by impaired glomerular filtration and protein loss. In these patients, early interventions aimed at controlling blood glucose and managing kidney function are crucial to slowing disease progression and preventing complications such as cardiovascular disease and anaemia.

The correlation between deep vein thrombosis (DVT) and lower creatinine levels is an intriguing finding [20]. Although DVT is typically associated with higher levels of creatinine due to impaired renal clearance, our results suggest that DVT may be linked to muscle catabolism from prolonged immobility, which may lower creatinine production, thus masking the true degree of renal dysfunction in these patients. This highlights the need to consider other factors, such as immobility and sarcopenia, when interpreting creatinine levels in patients with DVT [21,22].

Overall, these findings highlight that comorbidity–laboratory correlations are not merely statistical observations but carry direct implications for patient management. Smoking-related leukocytosis may prompt closer infection surveillance, alcohol-associated changes call for stricter hydration monitoring, and diabetic nephropathy markers support earlier nutritional and therapeutic interventions. Likewise, the paradoxical link between DVT and lower creatinine highlights the importance of integrating muscle mass and physical status into the routine evaluation of haemodialysis patients. Together, these insights argue for a personalised, comorbidity-driven interpretation of paraclinical tests in ESRD, with the ultimate goal of refining clinical decision-making and improving patient outcomes [23,24].

Our observation of sex-related differences in coagulation and cardiovascular biomarkers is consistent with the NHANES analysis, which showed distinct patterns of anaemia, mineral metabolism and cardiovascular complications in male versus female patients with chronic kidney disease.

In conclusion, our study depicts a multifaceted clinical profile of patients with advanced renal disease, characterised by mixed anaemia, coagulopathy and early cardiovascular involvement. Only modest sex-specific differences were observed—a prolonged APTT in men and higher platelet and D-dimer levels in women—which together emphasise the complexity of this high-risk population and the need for tailored clinical management.

### 4.4. Cross-Sectional Trends and Limitations

Correlations have been observed to converge towards an inflammatory–thrombotic profile in smokers and consumers of alcohol, a consistent pattern of renal dysfunction in diabetic and dialysis patients, and subtherapeutic INR values in most vascular pathologies. However, the validity of these interpretations is constrained by several factors. The lower median creatinine observed in the DVT group (3.0 mg/dL vs. 5.3 mg/dL) represents a cross-sectional association and should not be interpreted as a causal relationship.

The following recommendations are provided for further research:

The subsequent stage of the research should entail the verification of measurement units, the completion of any missing variables (collection date, body weight), the application of logistic and multivariate linear models with bootstrap for robust confidence intervals, the stratification of subjects by eGFR, and the external validation of the findings on an independent registry (2024–2025). In order to enhance the clinical relevance of these findings, it is essential that at least half of the associations present in an external dataset are replicated.

### 4.5. Study Limitations and Future Directions

This study has several limitations. The relatively small sample size (n = 50) and single-center retrospective design restrict the generalizability of our findings and reduce the statistical power of subgroup analyses. In addition, the cross-sectional nature of the study precludes causal inference. Although corrections for multiple testing and multivariate analyses were performed, the exploratory character of the study must be acknowledged. Another limitation is the lack of systematic data on inflammatory markers such as CRP, which may have provided further insight into thrombotic and bleeding risk. Despite these limitations, our study offers valuable preliminary insights into the complex interplay between comorbidities, anticoagulation, and paraclinical findings in haemodialysis patients. Future prospective multicenter studies with larger cohorts and extended biomarker profiling are warranted to validate and expand upon our results.

While this study provides important insights into the paraclinical profiles of CKD patients undergoing haemodialysis, several limitations must be acknowledged. First, the cross-sectional design of the study precludes the ability to establish causal relationships between comorbidities, anticoagulant therapy, and paraclinical findings. Long-term, prospective studies are required to clarify the temporal dynamics of these associations and to determine their sustained impact on patient outcomes. A further limitation of the present study is the relatively small sample size, particularly within certain subgroups—such as patients diagnosed with pulmonary embolism—which may have constrained the power of the analysis. Future studies should aim to include larger sample sizes to increase the generalizability of the findings. Finally, the study did not adjust for potential confounders such as age, sex, and medication use, which may have influenced the results. Multivariate analyses are needed to control for these variables and provide more robust conclusions.

## 5. Conclusions

This study provides evidence of the complex interplay between cardiovascular comorbidities, anticoagulant therapy, and paraclinical abnormalities in patients with ESRD on haemodialysis.

The findings confirm the heavy cardiovascular burden in this population and illustrate the delicate balance between preventing thromboembolic events and avoiding bleeding complications. Systematic monitoring of laboratory markers—such as coagulation profiles, haemoglobin, and D-dimer—is essential for optimizing patient management. Sex-specific differences observed in coagulation and cardiovascular biomarkers further support the need for individualized therapeutic strategies.

From a clinical perspective, our results highlight the importance of tailoring anticoagulation regimens to the individual risk profile of haemodialysis patients and of incorporating biomarkers such as D-dimer, BNP, and troponin into routine cardiovascular risk stratification.

Finally, while these findings provide valuable insights, they should be interpreted in the context of the study’s limitations. Future multicenter prospective studies with larger cohorts and extended biomarker profiling are warranted to validate and expand these observations. Ultimately, a multidisciplinary, comorbidity-driven approach remains crucial for improving both survival and quality of life in this high-risk group.

## Figures and Tables

**Figure 1 biomedicines-13-02387-f001:**
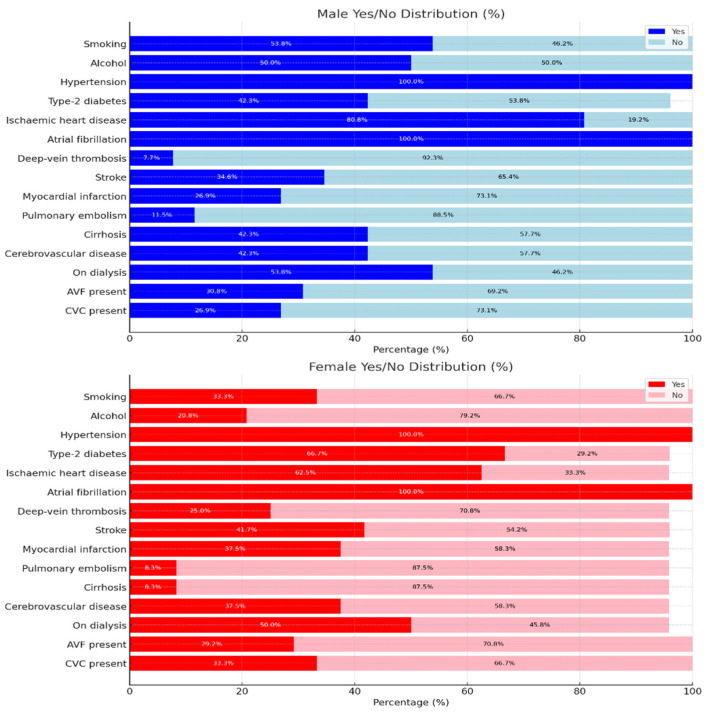
Gender-Based Health Condition Distribution (%). AVF = arteriovenous fistula; CVC = central venous catheter.

**Figure 2 biomedicines-13-02387-f002:**
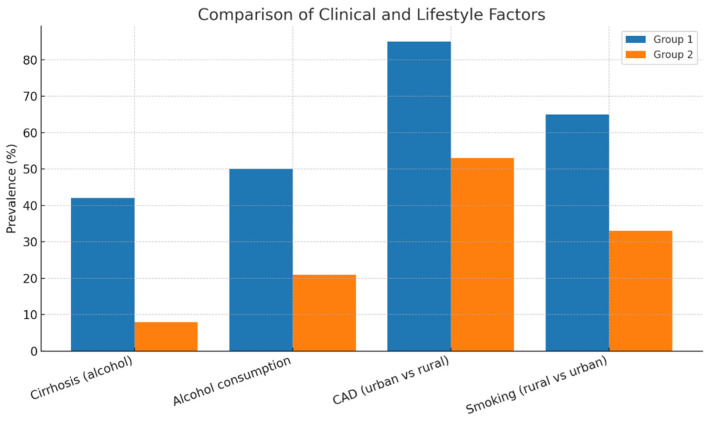
Comparative prevalence of cirrhosis, alcohol consumption, coronary artery disease, and smoking between patient subgroups. Group definitions varied according to the factor analyzed: cirrhosis and alcohol-consumers vs. non-consumers; coronary artery disease—urban vs. rural patients; smoking—rural vs. urban patients. CAD = coronary artery disease.

**Figure 3 biomedicines-13-02387-f003:**
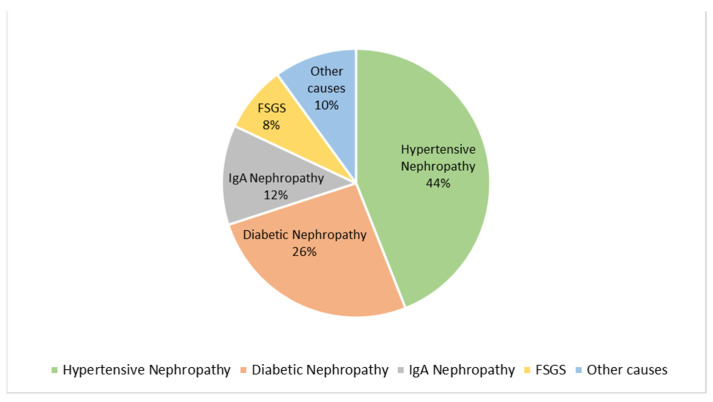
Baseline distribution of CKD etiologies among the first 50 patients included in the study. The most frequent cause was hypertensive nephropathy (44%), followed by diabetic nephropathy (26%), IgA nephropathy (12%), focal segmental glomerulosclerosis (FSGS, 8%), and other causes (10%).

**Table 1 biomedicines-13-02387-t001:** Prevalence of comorbidities in the study cohort.

	Male (n = 26)		Female (n = 24)	
Variable	Yes (%)	No (%)	Yes (%)	No (%)
Smoking	14 (53.8)	12 (46.2)	8 (33.3)	16 (66.7)
Alcohol consumption	13 (50.0)	13 (50.0)	5 (20.8)	19 (79.2)
Hypertension	26 (100.0)	0 (0.0)	24 (100.0)	0 (0.0)
Type-2 diabetes	11 (42.3)	14 (53.8)	16 (66.7)	7 (29.2)
Ischaemic heart disease	21 (80.8)	5 (19.2)	15 (62.5)	8 (33.3)
Atrial fibrillation	26 (100.0)	0 (0.0)	24 (100.0)	0 (0.0)
Deep-vein thrombosis	2 (7.7)	24 (92.3)	6 (25.0)	17 (70.8)
Stroke	9 (34.6)	17 (65.4)	10 (41.7)	13 (54.2)
Myocardial infarction	7 (26.9)	19 (73.1)	9 (37.5)	14 (58.3)
Pulmonary embolism	3 (11.5)	23 (88.5)	2 (8.3)	21 (87.5)
Liver Cirrhosis	**11 (42.3)**	**15 (57.7)**	**2 (8.3)**	**21 (87.5)**
Cerebrovascular disease	11 (42.3)	15 (57.7)	9 (37.5)	14 (58.3)
On dialysis	14 (53.8)	12 (46.2)	12 (50.0)	11 (45.8)
AVF present	8 (30.8)	18 (69.2)	7 (29.2)	17 (70.8)
CVC present	7 (26.9)	19 (73.1)	8 (33.3)	16 (66.7)

AVF = arteriovenous fistula; CVC = central venous catheter.

**Table 2 biomedicines-13-02387-t002:** Clinical and lifestyle characteristics by patient group (alcohol use, residence, and smoking status). *p*-values adjusted for multiple testing by the Benjamini–Hochberg FDR procedure.

Variable/Factor	Group 1	Group 2	Difference/Observation	*p*-Value	q (Benjamini–Hochberg Correction)
**Cirrhosis (regular alcohol consumption vs. no regular consumption)**	42%	8%	>5 times higher in consumers	0.01	0.03
**Alcohol consumption (yes vs. no)**	50%	21%	higher prevalence	0.01	0.03
**Complications at first contact with dialyzer (volume management/coagulation)**	-	-	trend toward increase	0.064	0.07
**Delayed AVF maturation**	-	-	increases exposure to bloodstream infections	-	0.05
**Residence (urban vs. rural)—timing of dialysis initiation**	-	-	no significant difference	0.71	0.07
**Coronary artery disease (urban vs. rural)**	85%	53%	higher in urban patients	0.021	0.04
**Smoking (rural vs. urban)**	65%	33%	higher in rural patients	0.034	0.05

AVF = arteriovenous fistula.

**Table 3 biomedicines-13-02387-t003:** Paraclinical findings in studied cohort.

Variable	Female (n)	Median ± SEM	Male (n)	Median ± SEM
Hemoglobin (Hb, g/dL)	22	9.70 ± 0.33	23	9.80 ± 0.35
Mean Corpuscular Volume (MCV, fL)	19	89.30 ± 1.28	22	91.30 ± 2.30
Hematocrit (% Hct)	22	30.15 ± 0.35	23	29.60 ± 0.99
Mean Corpuscular Hemoglobin Conc. (MCHC, g/dL)	22	33.20 ± 0.50	23	33.40 ± 0.42
Leukocytes (×10^3^/µL)	19	8.34 ± 1.74	23	7.27 ± 0.83
Lymphocytes (×10^3^/µL)	19	1.67 ± 0.53	22	1.22 ± 0.13
Neutrophils (×10^3^/µL)	19	5.08 ± 3.74	23	4.83 ± 0.78
Platelets (×10^3^/µL)	24	227.00 ± 15.08	25	191.50 ± 16.29
Creatinine (mg/dL)	24	4.83 ± 0.44	25	4.42 ± 0.67
Albumin (g/dL)	6	4.20 ± 0.43	8	5.80 ± 0.86
Total Protein (g/dL)	19	5.25 ± 0.30	21	5.60 ± 0.27
Sodium (Na, mmol/L)	24	135.00 ± 1.23	24	136.00 ± 0.83
Potassium (K, mmol/L)	23	4.20 ± 0.22	24	4.70 ± 0.14
RA (mmol/L)	22	21.00 ± 0.94	23	20.50 ± 0.60
Calcium (mg/dL)	19	8.60 ± 0.44	21	8.20 ± 0.36
Parathyroid Hormone (PTH, pg/mL)	–	–	7	42.00 ± 7.08
TSH (µIU/mL)	10	2.61 ± 1.02	5	1.51 ± 0.39
Serum Iron (µg/dL)	9	41.50 ± 13.01	10	47.00 ± 1.69
LDL-cholesterol (mg/dL)	1	101.6 ± 19.8	23	109.4 ± 24.7
HDL-cholesterol (mg/dL)	1	38.2 ± 6.1	24	34.5 ± 5.3
Total Cholesterol (mg/dL)	10	165.3 ± 25.4	18	174.6 ± 29.8
INR	24	1.89 ± 0.11	25	1.81 ± 0.32
APTT (s)	12	32.95 ± 5.90	11	41.60 ± 2.16
D-dimer (µg/mL)	9	3.14 ± 1.15	9	1.66 ± 1.13
BNP (pg/mL)	–	–	9	769.00 ± 229.60
CK-MB (ng/mL)	17	5.1 ± 2.1	11	17.18 ± 11.21
Troponin (ng/L)	7	45.2 ± 19.8	8	54.7 ± 24.5
TSH (µIU/mL)	10	2.61 ± 1.02	5	1.51 ± 0.39

RA = reserve alkaline; TSH = thyroid-stimulating hormone; LDL = low-density lipoprotein; HDL = high-density lipoprotein; INR = international normalized ratio; APTT = activated partial thromboplastin time; BNP = B-type natriuretic peptide; CK-MB = creatine kinase-MB isoenzyme.

**Table 4 biomedicines-13-02387-t004:** Associations between clinical factors and laboratory parameters with raw and FDR-adjusted *p*-values.

Compared Variables	Reported Values (Group 1 vs. Group 2)	Raw *p*-Value	FDR-Adjusted *p*-Value
Smokers vs. non-smokers—leukocyte count	8.8 vs. 7.0 × 10^3^/µL	0.030	0.060
Alcohol consumption—urine density	1.145 vs. 1.090	0.030	0.060
Type 2 diabetes—urea	145 vs. 118 mg/dL	0.027	0.054
Type 2 diabetes—total protein	5.0 vs. 5.75 g/dL	0.024	0.048
Deep vein thrombosis (DVT)—creatinine	3.0 vs. 5.33 mg/dL	0.001	0.005
Low INR—acute stroke & chronic cerebrovascular disease	1.52–1.60 vs. 2.03–2.07	0.025	0.050
Pulmonary embolism—HDL-C	7 vs. 18 mg/dL	0.009	0.022
Chronic dialysis—creatinine	5.69 vs. 3.83 mg/dL	0.004	0.013
Chronic dialysis—INR	1.51 vs. 2.11	0.001	0.005

DVT = deep vein thrombosis; INR = international normalized ratio; HDL-C = high-density lipoprotein cholesterol; FDR = false discovery rate.

## Data Availability

The data presented in this study are available upon request from the corresponding author.

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
