# Peer review of "The Interplay of Cardiovascular Comorbidities and Anticoagulation Therapy in ESRD Patients on Haemodialysis—The South-Eastern Romanian Experience"

_biomedicines, 2025, doi:10.3390/biomedicines13102387_

Round 1

Reviewer 1 Report

Comments and Suggestions for Authors

Congratulations to the authors for having produced  a manuscript about a populations which is often underrepresented in clinical studies; the manuscript contains many flaws that need to be corrected:

The manuscript describes the study as cross-sectional observational, yet the reported inclusion period (2021–2024) implies a longitudinal recruitment process. It should be clarified by the authors whether this was conducted as a retrospective chart review or as a prospective registry.

Laboratory units are inconsistently reported. For example, LDL and HDL values are implausibly low, suggesting potential unit errors. The authors should verify, standardize, and confirm all laboratory units.

Despite conducting more than 20 statistical comparisons, no correction for multiple testing was applied. Although the text acknowledges the risk of false discoveries, no adjustment was made. At a minimum, correction methods such as FDR or Bonferroni should be implemented by the authors.

Multivariate regression analyses are mentioned but not adequately presented in the results section; coefficients, odds ratios, and confidence intervals are missing. This analysis should be expanded and fully reported.

There are also inconsistencies between text and tables. For example, the prevalence of atrial fibrillation is variously described as 95–100%. These discrepancies should be resolved to ensure data accuracy.

Some laboratory results remain physiologically implausible (e.g., LDL reported at 10 mg/dL, troponin values) and must be carefully rechecked. The interpretation of creatinine levels in patients with deep vein thrombosis is speculative and should avoid causal inference unless supported by robust multivariate adjustment. Moreover authors are encouraged to include in their discussion other real world experiences of the use of DOACs (doi: 10.23736/S2724-5683.24.06546-3)

Author Response

Hello. please see the attachment. Thank you!

Reviewer 2 Report

Comments and Suggestions for Authors

The study addresses an important issue in nephrology, related to cardiovascular comorbidities in patients with chronic kidney disease (CKD) stage G5 on hemodialysis. However, I have several comments and a few ideas to strengthen the paper. I recommend the following revisions:

  • The title: the number of patients is small (50). So i suggest indicating the countary in the title.
  • Abstract: Background subsection : Kindly indicate the objective of the study; Methods subsection lacks details on parameters analyzed.
  • keywords need revision by citing main words. ESRD or CKD can be removed, "Thrombotic and bleeding risk" is not keyword. Kindly avoid using acronyms in keywords.
  • Introduction : The introduction states the purpose of the study. However, the hypothesis is neutral, Authors should present clear hypotheses about the relationships between comorbidities, anticoagulation therapy, and clinical outcome.
  • Materials and methods: This section should be restructured in subsections stduy design without indicating the objectives of the study; population, parameters, Statistical analysis, Inclusion and exclusion criteria should be defined.
  • Results section needs revision, there is a confusion between results and discussion. In this sections authors should just presnting the results without related them with factors or providing explaination.
  • acronyms used in tables and figures must be defined below them.
  • Discussion : The discussion provides a good interpretation of the results but lacks integration with relevant literature.
  • More references to previous studies supporting or contrasting the findings would strengthen the discussion.
  • A reflection on the limitations of the study, potential biases, and future research directions is necessary to strengthen the discussion.

  • The conclusion could be more impactful if it included concrete clinical recommendations based on the study's findings. It would also be relevant to add suggestions for future research.

Author Response

Hello. Please see the attachment. Thank you!

Reviewer 3 Report

Comments and Suggestions for Authors

This is an interesting and potentially useful study regarding the end stage renal disease. Although the study is relatively well designed, the conclusions are not entirely supported by the research methods and results.

  1. Being a statistical analysis, the number of patients taken into the study is rather small. This is acknowledged as a potential weakness by the authors but nevertheless it limits significantly the statistical power of the results.
  2. The "Materials and Methods" section is insufficiently developed. More detailed data about the patients should be added here, the primary cause of CKD should be specified (i.e., diabetes, high blood pressure, etc.) and the inconsistencies should be addressed (for instance, it is not clear if the starting age of the CKD patients was 18 or 30).
  3. The two groups of the patients should be specified in Figure 2 (what do they represent).
  4. The inflammatory status of the patients is not sufficiently analyzed. Additional investigations of the inflammatory markers should have been performed -CRP, immune status and different lymphocyte subpopulations distribution, oxidative stress markers, etc.
  5. The whole text (including the Abstract) should be checked for English language accuracy as there are syntax and typo errors.

Author Response

(The authors gave the same response as above.)

Round 2

Reviewer 1 Report

Comments and Suggestions for Authors

congratulations to the authors for the revised version of their manuscript.

Author Response

Thank you very much for your kind and constructive feedback on my manuscript. I am pleased to hear that the revisions have addressed all your concerns. Your comments and guidance have been invaluable in improving the quality of my work, and I truly appreciate the time and effort you have dedicated to reviewing it.

I am confident that your input will greatly support the publication of this article, and I am sincerely grateful for your contribution to this process.

Best regards,
Dr. Suliman Ioana Livia

Reviewer 2 Report

Comments and Suggestions for Authors

The authors have provided a nicely detailed and thorough response to the comments from the previous review and have addressed my major concerns. However, some responses need additional details, and I have additional comments for the author to consider:

- I suggest revising the title by citing the origin of the patients (the findings cannot be generalized).

- In the abstract, Kindly add a clear objective of the study.

- Methods subsection lacks details on parameters evaluated; “clinical and paraclinical data” still unclear.

- Keywords the origin of patients should be indicated.  

-  Methodology

The subsection data collected and parameters

Authors consider specifying the parameters evaluated

-  Results

Some acronyms ues should be defined (LMWH, NOACs..)

Some statements should be placed in the Discussion section, example : “This finding may reflect greater access to medical services, an older biological age”

Author Response

Hello. Please see the attachment. 

Reviewer 3 Report

Comments and Suggestions for Authors

The issues raised during the first revision of the paper were addressed properly and suitable explanations were provided for the concerns addressed, especially those related to some missing laboratory data. English language was improved.

Author Response

(The authors gave the same response as above.)

Round 3

Reviewer 2 Report

Comments and Suggestions for Authors

The authors have provided a nicely detailed and thorough response to the comments from the previous review and have addressed my major concerns. However, one response needs additional details and I have additional comment for the author to consider and:

- Cross-sectional trends and limitations subsection must be place in teh discussion section.
- Some statements should be placed in the Discussion section, example:Collectively, these findings delineate a picture of advanced renal dis-
ease complicated by mixed anaemia, coagulopathy and incipient cardiovascular involve-
ment, with only modest sex-specific divergences—namely, a longer APTT in men and
higher platelet and D-dimer levels in women.

- Please ensure that the figures above do not include titles.

Author Response

We would like to sincerely thank the reviewers for their thorough evaluation of our manuscript and for their constructive and helpful comments. We have carefully revised the manuscript in accordance with all suggestions. Below we provide a point-by-point response to each comment.

Reviewer’s comment 1:

Please add a subsection on “Cross-sectional Trends and Limitations” in the Discussion section. Some statements should be placed in the Discussion section (e.g. “Collectively, these findings…”).

- We thank the reviewer for this valuable suggestion. We have created a dedicated subsection entitled “Cross-sectional Trends and Limitations” in the Discussion section and have moved the summary statements, including the sentence previously in the Results (“Collectively, these findings…”), into the Discussion section as recommended.

Reviewer’s comment 2:

Please ensure that figures do not contain titles.

- We have carefully checked all figures and removed any titles from within the figure panels to comply with the journal’s format requirements.